# MR-SPLIT: A novel method to address selection and weak instrument bias in one-sample Mendelian randomization studies

**Ruxin Shi[1], Ling Wang[2], Stephen Burgess[3], Yuehua Cui[1]***

**1** Department of Statistics and Probability, Michigan State University, East Lansing, Michigan, United States of America, **2** Department of Medicine, Michigan State University, East Lansing, Michigan, United States of America, **3** Biostatistics Unit, University of Cambridge, Cambridge, United Kingdom

\* cuiy@msu.edu

## Abstract

Mendelian Randomization (MR) is a widely embraced approach to assess causality in epidemiological studies. Two-stage least squares (2SLS) method is a predominant technique in MR analysis. However, it can lead to biased estimates when instrumental variables (IVs) are weak. Moreover, the issue of the winner's curse could emerge when utilizing the same dataset for both IV selection and causal effect estimation, leading to biased estimates of causal effects and high false positives. Focusing on one-sample MR analysis, this paper introduces a novel method termed Mendelian Randomization with adaptive Sample-sPLitting with cross-fitting InstrumenTs (MR-SPLIT), designed to address bias issues due to IV selection and weak IVs, under the 2SLS IV regression framework. We show that the MR-SPLIT estimator is more efficient than its counterpart cross-fitting MR (CFMR) estimator. Additionally, we introduce a multiple sample-splitting technique to enhance the robustness of the method. We conduct extensive simulation studies to compare the performance of our method with its counterparts. The results underscored its superiority in bias reduction, effective type I error control, and increased power. We further demonstrate its utility through the application of a real-world dataset. Our study underscores the importance of addressing bias issues due to IV selection and weak IVs in one-sample MR analyses and provides a robust solution to the challenge.

## Author summary

Mendelian randomization (MR) is a method used in genetic epidemiology to determine whether a specific exposure has a causal effect on a health outcome. Ensuring the accuracy of this method is crucial for its reliability and for making informed decisions that can enhance public health and medical practices. Typically, researchers employ the two-stage least squares (2SLS) method which involves selecting a set of valid instrumental variables (IVs) to estimate and infer the causal effect. However, 2SLS can produce biased results when the effects of the IVs are weak, known as weak instrument bias. Additionally, the "winner's curse" problem may occur when using the same dataset for both IV selection

**Data Availability Statement:** The Chronic Renal Insufficiency Cohort (CRIC) Study was conducted by the CRIC Study Investigators and supported by

the National Institute of Diabetes and Digestive and Kidney Diseases (NIDDK). The data from the CRIC Study reported here were supplied by the CRIC investigators and were downloaded from dbGaP with accession number phs000524.v1.p1. This manuscript was not prepared in collaboration with Investigators of the CRIC Study and does not necessarily reflect the opinions or views of the CRIC Study, or the NIDDK. R codes for MR-SPLIT can be freely accessed at: https://github.com/RuxinShi/MRSPLIT.

**Funding:** This work was supported by a grant from the American Heart Association (24TPA1288424 to LW and YC). The funder had no role in study design, data collection and analysis, decision to publish, or preparation of the manuscript.

**Competing interests:** The authors have declared that no competing interests exist.

and causal effect estimation, introducing additional bias. Here we introduce a novel approach called MR-SPLIT, which addresses these two bias issues by randomly splitting the data into two parts: one for IV selection and the other for IV construction and causal effect estimation. Through effective integration, this strategy enhances the power, reduces bias, and provides more precise estimates. Our approach is validated through extensive simulation studies, and its effectiveness is demonstrated by an application to a real-world dataset.

## Introduction

Mendelian Randomization (MR) utilizes genetic variants as instruments to detect causal effects between an exposure and an outcome variable [1–4], and has emerged as a pivotal method in the realm of causal inference. In observational studies, establishing a causal relationship between two variables exhibiting an observed association can be challenging, because of unknown confounding factors that may influence both variables simultaneously. However, MR harnesses Single Nucleotide Polymorphisms (SNPs) as instrumental variables (IVs) to infer causative associations between exposures and outcomes, circumventing the confounding biases inherent in conventional observational studies. Given that SNPs, as genetic variants, are randomly segregated during meiosis, they offer a unique opportunity for robust causal inference.

This innovative utilization of SNPs as IVs rests on a triad of fundamental assumptions, which are indispensable for ensuring the validity of MR results [4, 5]: 1) Relevance assumption; 2) Independence assumption; and 3) Exclusion-restriction assumption. Using SNPs as IVs can also avoid reverse causation because genetic variants are randomly assigned at conception and remain constant throughout an individual's life. However, violations [6] of the triadic assumptions underpinning MR analysis can produce biased and unreliable estimates. Specifically, when the relevance assumption is contravened, we encounter what is termed a "weak instrument" [7, 8] phenomenon. Conversely, breaches of the latter two assumptions often manifest as "pleiotropy" [9–11], where the instrumental variable exerts effects on the outcome via pathways other than the exposure of interest. Throughout this work, the primary focus is to address the challenge of IV selection and weak instrument bias in one-sample MR analyses.

In MR analysis, two main frameworks are commonly used: two-sample MR analysis with GWAS summary statistics and one-sample MR analysis with individual-level data. While two-sample MR analysis has gained popularity due to easy access to public datasets, it comes with a couple of limitations. Firstly, it relies on marginal estimates of SNP statistics, which can be biased when not accounting for linkage disequilibrium (LD) properly. Secondly, it lacks the flexibility to model other causal mechanisms, such as nonlinear causal effects. As a result, there continues to be a significant interest in the advancement of statistical methods for one-sample MR analysis. The most popular method used in one-sample MR analysis is the two-stage least squares (2SLS) approach [12], which is relatively straightforward to implement and can yield consistent estimates of causal effects. However, the 2SLS estimate can be biased in the presence of weak instruments [8]. The bias is in the direction of the confounded association and can cause inflated false positive rates, particularly when more than one IV is included in the analysis [13]. To date, weak instrument bias still remains one of the significant concerns in one-sample MR analysis [13].

A potential solution to mitigate the impact of weak IVs is to opt for a two-sample MR analysis. While this approach might mitigate some biases, it does not eliminate them entirely.

Specifically, bias due to weak instruments in two-sample MR tends to be directed towards the null [14]. Limited information maximum likelihood (LIML) method [15, 16] was introduced as an alternative to 2SLS when dealing with weak instruments. Burgess et al. [17] showed that LIML could provide a less biased estimate compared to 2SLS in the presence of weak instruments, but at the expense of incurring larger variance. Nevertheless, LIML is still subject to weak instrument problems and its finite sample performance can be poor. Angrist et al. [18] proposed two jackknife instrumental variables estimators (JIVE) as alternatives to 2SLS and LIML to reduce the bias with many weak instruments. However, Sören and Matz [19] showed that neither LIML nor the JIVE estimators perform uniformly better than the 2SLS does in terms of root mean square error.

In one-sample MR analysis, when the same dataset is used for both IV selection and causal effect estimation, the "winner's curse" or IV selection bias emerges as another notable concern in addition to the weak IV bias issue [13, 20]. This bias could lead to biased causal effect estimates and hence inflate false positive rates under the 2SLS IV regression framework. This is evident in Section 2 of the S1 Text, where it is shown that using the same data (the whole sample) for both IV selection and causal effect estimation, both LIML and 2SLS methods amplify bias compared to using half data for selection and the other half for causal effect estimation. Thus, it is critical to address the IV selection bias issue in one-sample MR analysis. Typically, in one-sample MR analysis, IVs are typically chosen based on a p-value threshold. However, the usage of a p-value threshold criterion in the selection of IVs is somewhat arbitrary and lacks robust justification. The 2SLS approach relies on the fitted values from the first stage for estimating causal effects in the second stage, highlighting the critical role of prediction accuracy and thus questioning the robustness of models that depend solely on p-value thresholds for validation. Given the typically vast dimensionality of SNP data, the use of penalized shrinkage methods can effectively mitigate the winner's curse effect in one-sample MR analysis. This strategy prioritizes prediction accuracy and hence provides a potentially more dependable and robust framework for causal inference.

Denault et al. [21] introduced a method called 'Cross-Fitting for Mendelian Randomization' (CFMR) to handle the weak instrument issue in one-sample MR analysis, which consolidates information from multiple IVs into a single IV, termed the Cross-Fitted Instrument (CFI). CFMR randomly splits a sample into $K$ subgroups $\{I_1, \cdots, I_K\}$ and define the complement of the partition $I_k$ as $I_k^c = \{1, \cdots, N \notin I_k\}$. In each subset $\{I_k^c\}$, it first selects $\gamma_k$ independent variants $\{Z_{1,k}, \cdots, Z_{\gamma_k,k}\}$, and then defines a CFI of the exposure $X$ on $I_k$, which is the prediction of $X$ on $I_k$ trained using data with indexes in $I_k^c$.

This predicted value can be viewed as a polygenic risk score in risk prediction analysis. Then, the new CFI is used as the IV to fit the 2SLS model for further causal inference. CFMR consolidates all the IVs into one single IV (CFI), thus it produces less biased results. However, CFMR does not completely solve the selection bias issue. Taking $K = 10$ as an example, CFMR employs 9 folds of data for selecting IVs and applies the estimated effects to construct the composite IV in a separate fold of data. By iterating this process 10 times, the composite IVs across any pair of folds are constructed with 80% of data in common. Thus, the composite IVs are not constructed using completely independent data. This could lead to a new manifestation of the winner's curse problem. Furthermore, by relying on one CFI as the only IV to represent the collective information of all IVs, there could be potential information loss which further leads to variance inflation and consequently reduced power (as shown in our theorem and simulation studies).

In general, the selection of IVs involves a bias and variance trade-off when estimating the causal effect. Using more IVs tends to introduce a larger bias but smaller variance, whereas

employing too few IVs results in a smaller bias but larger variance. Pierce et al. [22] did intensive simulations to evaluate the power and IV strength requirements for MR analyses based on 2SLS. They employed four strategies to combine information across IVs and evaluated the consequences of these strategies on power and overall IV strength, as measured by the first-stage F statistic in 2SLS. The results suggest that categorizing IVs into major and weak ones and then consolidating the weak ones into a single IV based on the knowledge of the genetic architecture underlying the exposure, can mitigate the issue of weak IVs. However, the study identifies a gap in current methodologies: it does not provide a clear approach for differentiating between major and weak IVs, nor does it offer a strategy for combining weak IVs in the context of one-sample MR analysis. This highlights an area for further research and methodology development in the field.

In this work, we propose an adaptive Sample-sPLitting method with cross-fitting InstrumenTs (MR-SPLIT) to address the bias issue of IV selection and weak instruments. This approach can effectively reduce the number of weak IVs without the loss of much information, thereby enhancing the performance of causal inference in MR studies by improving the power of causal inference. Our method has two advantages over the existing ones: 1) It adaptively selects major and weak IVs, subsequently creating a composite IV from the weaker ones. We theoretically proved that the variance of the MR-SPLIT estimate is smaller than that of the CFMR estimate under the condition of one sample split. Simulation results also show that MR-SPLIT can always achieve higher power and lower RMSE than CFMR; and 2) A multi-sample splitting strategy is further employed to enhance the robustness of estimation and testing. Extensive simulation studies were conducted to assess the performance of our method in comparison to its counterparts, including 2SLS, LIML, and CFMR. Our method offers an efficient and powerful solution for one-sample MR analysis by addressing two primary sources of bias: IV selection bias and the bias associated with weak instruments.

## Statistical method

Assume the following structural equation model,

$$y_i = x_i \beta + \varepsilon_{yi}$$

$$x_i = G_i \alpha + \varepsilon_{xi}$$

where $x_i$ is the exposure, and $y_i$ denotes the outcome of the $i$th individual. $G_{i.}$ is a $p$-dim vector of SNP IVs, where $G_{i.} = \{G_{i1}, G_{i1}, \ldots, G_{ip}\} \in \mathbb{R}^p$. The error term is denoted by $\varepsilon_i = (\varepsilon_{xi}, \varepsilon_{yi}) \sim N(0, \sigma^2 R)$ where $R_{12}(= \rho)$ is the correlation due to confounding. $\beta$ is the interested causal effect which needs to be estimated. Suppose we have $N$ independent individuals, and denote $Y = (y_1, \ldots, y_N)' \in \mathbb{R}^{N \times 1}, X = (x_1, \ldots, x_N)' \in \mathbb{R}^{N \times 1}, G = \{G_1, \ldots, G_p\} \in \mathbb{R}^{N \times p}$, where the $j$th IV denoted as $G_j = (G_{1j}, \ldots, G_{Nj})', j = 1, \ldots, p$, then we have

$$Y = X\beta + \varepsilon_y$$

$$X = G\alpha + \varepsilon_x$$

(1)

## MR-SPLIT: Mendelian Randomization with adaptive Sample-sPLitting with cross-fitting InstrumenTs

**Cross-fitting instruments with sample split.** Given the observed data $\{X, Y, G\}$, we first need to select a valid IV subset from the existing SNP pool, where the number of SNPs can be

much larger than the sample size. To reduce potential biases and enhance the accuracy of estimates in MR analysis, one can use one sample for the selection of appropriate IVs and a separate, independent sample for the 2SLS estimation. By doing so, over-fitting and biases stemming from sample-specific peculiarities, such as the double dipping issue, can be minimized, leading to more robust and credible causal effect estimates. When only one sample is available, one simple idea is to randomly split the data into two equal subsets $\{I_1, I_2\}$, each containing roughly $N/2$ samples. Then, one can use one subset (say $I_1$) to select the IVs and use the other (say $I_2 = I_1^c$) to get the estimates of $\beta$.

For the IV selection, if no prior information about specific SNPs is available, researchers usually regress the exposure variable on each SNP, and then select those SNPs that yield marginal p-values smaller than a preset threshold (e.g., $5 \times 10^{-8}$) followed by LD pruning or LD clumping. However, such a threshold is quite ad hoc and sometimes can be too stringent, prompting the need for relaxation, as advocated in some studies [23]. Such strictness can lead to the exclusion of valid IVs and the loss of valuable information. Conversely, if the threshold is too lenient, it may result in the selection of an excessive number of SNP IVs, potentially introducing challenges associated with weak IVs [17]. We suggest using some high-dimensional screening methods such as sure independence screening (SIS) [24] to first reduce the SNP dimension from ultra-high to high dimension. Methods like SIS have the sure screening property in which they ensure that, as the sample size increases, the probability of including all relevant variables becomes close to one. After this step, shrinkage methods such as LASSO or adaptive LASSO [25, 26] can be employed to select and estimate non-zero SNP effects. Other penalized methods with different penalty functions such as MCP or SCAD can also be applied.

After the SNP selection, directly employing these IVs in 2SLS might lead to the issue of weak instruments, potentially resulting in biased estimate. To mitigate this, we group the IVs into two groups, major IVs and weak IVs, based on their association strength with the exposure. Conventionally, the validity of IVs is assessed using $F$ statistics. A common benchmark used in econometrics and statistical literature suggests that an F-statistic exceeding 10 is indicative of strong instruments, particularly when assessing the strength of a collective set of IVs [27, 28]. However, the determination of the weakness of an individual IV lacks a widely recognized standard. In this analysis, we employed partial F-statistics with different thresholds as criteria for selecting major IVs. Generally, the partial F statistic is defined as:

$$F = \frac{(\mathrm{RSS}_r - \mathrm{RSS}_f)/p}{(\mathrm{RSS}_f)/(N - k - 1)} \tag{3}$$

where $\mathrm{RSS}_r$ and $\mathrm{RSS}_f$ are the residual sums of squares for the reduced and full model, respectively; $N$ is the total number of observations; $k$ and $p$ are the numbers of variables in the reduced and full model, respectively. This statistic measures how much the addition of $p$ variables improves the model, compared to the increase in complexity these variables bring. It is a good statistic for calculating the strength of each IV and is consistent with the commonly used F-statistic for evaluating IV strength. In our model, $p = 1$ because we calculate the partial F statistics for each IV. The thresholds were set at partial F-statistics greater than 10, 30, and 50. We conducted a simulation study to compare these three statistics for the purpose of identifying weak IVs, as detailed in section Evaluation of Major IV identification. Based on the simulation results, it is recommended to use a threshold of partial $F > 30$ to define major IVs.

**Consolidating weak IVs to form a composite IV.** Following the separation of major and weak IVs, we propose consolidating the weak ones into a composite IV. Then, the major IVs and the composite IV are included in the 2SLS model to infer the causal effect (see Fig 1 for the flowchart of MR-SPLIT). By only consolidating the weak IVs into a single instrument, we

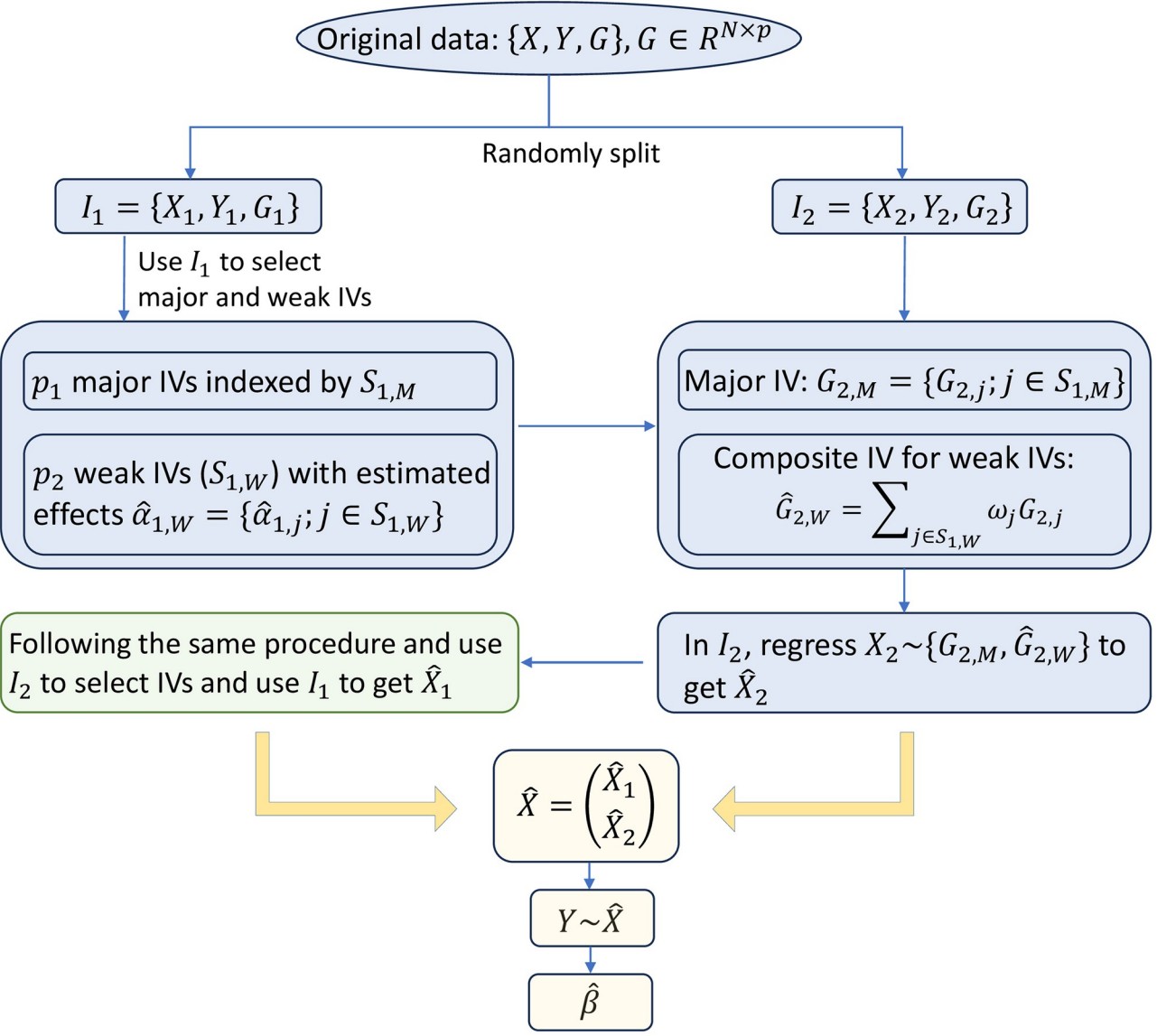

**Fig 1. The flow chart of MR-SPLIT with one random split.** The original data is randomly split into two parts indexed by $I_1$ and $I_2$. We use data in $I_1$ to select major and weak IVs, then form the composite IV for the weak ones in $I_2$ with the weight $\omega$ being calculated based on Eq (3), then get the fitted $\hat{X}_2$ in $I_2$. Similarly, we use data in $I_2$ to select major and weak IVs, then use data in $I_1$ to form the composite IV and get the fitted values $\hat{X}_1$. Next, we combine $\hat{X}_1$ and $\hat{X}_2$ to get $\hat{X} = (\hat{X}_1^T, \hat{X}_2^T)^T$ and fit the 2nd stage regression model $Y \sim \hat{X}$ to get the causal estimate ($\hat{\beta}$) and its testing p-value.

can substantially reduce the number of IVs in the model while retaining most of the information they carry.

Denote the selected index of weak IVs as $S_{k,W}$, $k = 1, 2$, where $|S_{k,W}| = p_2$ represents the selected numbers of weak IV using data in $I_k$. Taking sample $I_1$ as an example, let the estimated effects for the weak IVs on the exposure be denoted as $\hat{\alpha}_{1,W} = \{\hat{\alpha}_{1,j}; j \in S_{1,W}\}$ for data in $I_1$. Here, the subscript 1 indicates that this parameter is estimated from subsample $I_1$, and the subscript $W$ signifies that it corresponds to the direct effect of weak IVs on $X$ from Eq (1). The

new composite IV constructed in sample $I_2$, $\hat{G}_{2,W}$, is then defined as

$$\hat{G}_{2,W} = \sum_{j \in S_{1,W}} \omega_j G_{2,j} \tag{2}$$

where

$$\omega_j = sign(\hat{\alpha}_{1,j}) \frac{|\hat{\alpha}_{1,j}|}{\sum_{j \in S_{1,W}} |\hat{\alpha}_{1,j}|} \tag{3}$$

In other word, we use weak IVs selected from sample $I_1$ to construct the new composite IV in sample $I_2$. Then, we can use the major IVs and the new composite IV, i.e., $\{G_{2,M}, \hat{G}_{2,W}\}$, to get the cross-fitted exposure in $I_2$. Here, the subscript $M$ represents that $G_{2,M}$ is identified as the major IV, while the subscript $W$ signifies that $\hat{G}_{2,W}$ is estimated from the weak IV. The subscript 2 indicates that these values are obtained from subsample $I_2$.

To clarify logic, for each $k = 1, 2$ we use the subset $I_k$ to identify the SNP IVs and obtain the estimated effects $\hat{\alpha}$ for the selected IVs. Then, we categorize them into two groups, major IVs and weak IVs, using the partial $F$-statistic criterion defined earlier. We then combine the weak IVs in $I_k^c$ using the estimated weights from $I_k$. This approach enables us to avoid overfitting by selecting IVs and estimating the causal effect using different samples.

**Estimating the causal effect.**   Once we get the IVs $\{G_M, \hat{G}_W\}$ in each $I_k$, $k = 1, 2$, we can then perform the first stage of the 2SLS regression on these IVs to get the cross-fitted exposures $\hat{X}_k$ which are then aggregated, i.e.,

$$\hat{X} = \begin{pmatrix} \hat{X}_1 \\ \hat{X}_2 \end{pmatrix} \in \mathbb{R}^{N \times 1}.$$

The causal effect can be estimated by regressing $Y$ on $\hat{X}$ using the whole sample, which is given by

$$\hat{\beta} = (\hat{X}'\hat{X})^{-1} \hat{X}'Y \tag{4}$$

Cross-fitting allows for the utilization of the entire dataset in estimating causal effects, thereby circumventing the winner's curse problem that arises when the same data is employed for both IV selection and causal effect estimation.

**Remark 1:** Both MR-SPLIT and CFMR implement a cross-fitting idea with sample splitting, but the analysis is fundamentally different. MR-SPLIT combines the cross-fitted exposures for further causal inference, while CFMR combines cross-fitting instruments for further causal inference. CFMR first calculates the composite IV in the $k$th split sample (denoted as $\tilde{G}_{k,n_k \times 1}$), where $n_k$ denotes the sample size in the $k$th split sample, then combines these composite IVs to form the final composite IV (denoted as $\tilde{G}_{N \times 1}$ by stacking all $\tilde{G}_k$), and finally uses the full data $(X, \tilde{G}, Y)$ to perform 2SLS analysis for causal inference. Each composite IV $\tilde{G}_{k,n_k \times 1}$ can be regarded as a polygenic risk score based on the $k$th split sample. As the dimensions of major and weak IVs identified from different sample splits are different, CFMR is infeasible to separate the two components and incorporate them in downstream causal inference.

**Remark 2:** CFMR uses data in $I_k^c$ to select IVs, then calculates the composite IV based on data in $I_k$. Typically, a 10-fold split suffices. On the other hand, MR-SPLIT benefits from a 2-fold sample splitting. It uses data in $I_1$ to select and separate major and weak IVs, then forms the cross-fitting composite IV in $I_2$. After that, it calculates the cross-fitted exposures based on

the major IV(s) and the cross-fitting composite IV for further causal inference. More data for IV selection leads to less data to fit the cross-fitted exposure and vice versa. To balance the two components, a 2-fold sample split is recommended.

**Remark 3:** By combining the cross-fitted exposures, Theorem 1 shows that MR-SPLIT produces estimates with a variance no larger than that of CFMR if both approaches implement a 2-fold sample split. The proof is given in S1 Text. This finding extends to scenarios involving a $k(>2)$-fold sample split for CFMR. Although providing theoretical proof for this result poses a challenge, we have demonstrated its validity through simulations.

**Theorem 1** *Let $\hat{\beta}_{CFMR}$ and $\hat{\beta}_{MR-SPLIT}$ be the 2SLS estimates obtained respectively by the CFMR and MR-SPLIT method with a 2-fold random split. Then, $\hat{\beta}_{MR-SPLIT}$ is more efficient than $\hat{\beta}_{CFMR}$ in the sense that*

$$var(\hat{\beta}_{MR-SPLIT}) \leq var(\hat{\beta}_{CFMR})$$

The proof of Theorem 1 is given in S1 Text.

**Multiple sample splitting to improve robustness.** Given the inherent uncertainty in single-sample splitting, particularly in cases of limited sample size, we propose a multiple-splitting strategy to improve the robustness of the approach. We randomly split data (into two halves) $L$ times. For each random split, the same estimation and testing procedure as described before are conducted. Let $pval_l$ denote the p-value at the $l$th random split. There are different ways to aggregate these $L$ p-values. One approach involves employing the aggregation method for p-values proposed by Wasserman and Roeder [29, 30]. However, this method has proven to be overly conservative in our simulations. Another simple way is to use the Cauchy combination rule for correlated p-values [31], which is similar to the minimum p-value method but does not require an intensive resampling procedure to assess the null distribution of the minimum p-value. Given its computational efficiency, we adopt the Cauchy combination rule to aggregate p-values obtained from multiple sample splitting. Following [31], the test statistics is defined as:

$$T_{cauchy} = \sum_{l=1}^{L} \omega_l \tan((0.5 - pval_l)\pi)$$

where the weights $\omega_l$ are non-negative and $\sum_{l=1}^{L} \omega_l = 1$. If no further information is available, the weight $\omega_l$ can be simply chosen as $1/L$. The p-value of $T_{cauchy}$ can be simply approximated by

$$\text{p-value} = \frac{1}{2} - \arctan(T_{cauchy})/\pi \tag{5}$$

In essence, augmenting the number of sample splits improves result robustness. However, this enhancement comes with the trade-off of requiring increased computational resources. To provide general guidance on the number of splitting times, we conducted a simulation study (see section Simulation study). The results suggest that conducting multiple splits about 50–60 times is sufficient to achieve a robust outcome in terms of controlling type I errors and maintaining stable statistical power. In the case of a large sample size and strong SNP heritability, the splitting time can be dramatically reduced (see the simulation results).

**Algorithm**
The detailed algorithm of the MR-SPLIT is given below:

1. For each $l = 1, \cdots, L$ random split, repeat the following steps:

a. Split the sample into two equal subsets $\{I_1, I_2\}$, i.e., $\{1, \cdots, N\} = I_1 \cup I_2$ with $I_1 \cap I_2 = \emptyset$ and $|I_1| = [N/2]$ and $|I_2| = N - [N/2]$, and denote the complementary sets as $\{I_1^c, I_2^c\}$ accordingly.

b. For each $k = 1, 2$, we use $I_k^c$ to select IVs and get the estimated effect size for each IV. Then, categorize the selected IVs into two distinct groups, major IV(s) and weak IVs, based on the partial $F > 30$ criterion.

c. In each subset $I_k$, combine the weak IVs using the effect size estimated from $I_k^c$ following Eq (2). Then regress the exposure variable $X$ on the new IVs (major IV(s) + composite IV) to get the fitted value $\hat{X}_k$.

d. Denote $\hat{X} = \begin{pmatrix} \hat{X}_1 \\ \hat{X}_2 \end{pmatrix}$, and do the second stage regression of $Y$ on $\hat{X}$ to get the causal effect estimate $\hat{\beta}_l$ and the p-value $pval_l$.

2. Calculate the Cauchy combination statistics $T_{\text{cauchy}} = \frac{1}{L} \sum_{l=1}^{L} \tan((0.5 - pval_l)\pi)$, and the aggregated p-value as $pval = \frac{1}{2} - \arctan(T_{cauchy})/\pi$. The final aggregated causal effect estimate can be calculated as $\hat{\beta} = \frac{1}{L} \sum_{l=1}^{L} \hat{\beta}_l$.

## Results

### Simulation study

We conducted simulations to assess the performance of our method and provided guidance on the identification of major IVs and selecting an efficient number of sample splitting. Subsequently, we compared the proposed MR-SPLIT with the existing approaches, including 2SLS, LIML, and CFMR, across various settings.

**Evaluation of Major IV identification.** We applied 3 criteria, $F > 10$, $F > 30$, and $F > 50$, to distinguish the major and weak IVs under various settings. We randomly generated 300 independent SNPs each with MAF = 0.3, and assumed only 5 SNP had effects on the exposure. The effects of these SNPs were set to be $\beta = (0.4, 0.4, 0.1, 0.05, 0.05)\omega_0$, where $\omega_0$ was chosen to ensure that these SNPs account for $h^2 = \{0.15, 0.30, 0.50\}$ of the variation in exposure ($h^2$ can be interpreted as the exposure heritability). The error term was assumed to follow the standard normal distribution with mean 0 and variance 1. The rest 295 SNPs were assumed to be noises with no effect on the exposure (i.e., $\beta = 0$). Then, we followed model (1) to simulate the exposure. In this setting, the initial two SNPs may be regarded as the major IVs, whereas the remaining three are categorized as weaker ones. However, this differentiation can also be contingent on the signal-to-noise ratio, meaning that the first two SNPs may not be deemed as the major ones when $h^2$ is low, say $h^2 = 0.15$. And when the IVs are strong enough, say $h^2 = 0.5$, the three weaker IVs may be regarded as strong IVs. After applying SIS screening and LASSO estimation on these 300 SNPs, we then used these three criteria to distinguish the major and weak IVs. Part of the results can be seen in Table 1. As we mentioned before, there were 295 noise SNPs in total. It is possible that some of these noise SNPs may be incorrectly identified as major IVs. We also summarized these results in the last column of Table 1. More detailed information can be found in Table A and Fig A in S1 Text. Our analysis indicates that employing a partial $F > 10$ threshold to define major IVs is excessively lenient, leading to misidentifying noises as major IVs, particularly in scenarios with small sample sizes (e.g., $N = 500$). Conversely, a threshold of $F > 50$ proves overly stringent, failing

**Table 1. Mean numbers of being identified as major IV using different criteria in 1,000 simulations.**

| $h^2$ | $N$ | Criteria | $SNP_1$ | $SNP_2$ | $SNP_3$ | $SNP_4$ | $SNP_5$ | Noises (×295)* |
|---|---|---|---|---|---|---|---|---|
| 0.15 | 500 | F>10 | 0.55 | 0.5 | 0.15 | 0 | 0.1 | 1.25 |
| | | F>30 | 0.05 | 0.05 | 0 | 0 | 0 | 0 |
| | | F>50 | 0 | 0 | 0 | 0 | 0 | 0 |
| | 1000 | F>10 | 0.95 | 0.95 | 0.35 | 0.1 | 0 | 0.65 |
| | | F>30 | 0.35 | 0.35 | 0 | 0 | 0 | 0 |
| | | F>50 | 0.15 | 0 | 0 | 0 | 0 | 0 |
| | 2000 | F>10 | 1 | 1 | 0.75 | 0.35 | 0.55 | 0.6 |
| | | F>30 | 1 | 0.95 | 0.1 | 0 | 0 | 0 |
| | | F>50 | 0.75 | 0.75 | 0 | 0 | 0 | 0 |
| 0.3 | 500 | F>10 | 0.95 | 0.8 | 0.25 | 0.25 | 0.1 | 1.15 |
| | | F>30 | 0.5 | 0.55 | 0 | 0 | 0 | 0 |
| | | F>50 | 0.25 | 0.25 | 0 | 0 | 0 | 0 |
| | 1000 | F>10 | 1 | 1 | 0.75 | 0.45 | 0.3 | 0.65 |
| | | F>30 | 1 | 1 | 0.2 | 0 | 0 | 0 |
| | | F>50 | 0.8 | 0.7 | 0.05 | 0 | 0 | 0 |
| | 2000 | F>10 | 1 | 1 | 1 | 0.75 | 0.9 | 0.6 |
| | | F>30 | 1 | 1 | 0.6 | 0.15 | 0.1 | 0 |
| | | F>50 | 1 | 1 | 0.1 | 0 | 0.05 | 0 |
| 0.5 | 500 | F>10 | 1 | 1 | 0.8 | 0.35 | 0.4 | 1.8 |
| | | F>30 | 1 | 1 | 0.35 | 0 | 0.05 | 0 |
| | | F>50 | 0.9 | 0.9 | 0 | 0 | 0 | 0 |
| | 1000 | F>10 | 1 | 1 | 1 | 1 | 0.95 | 0.6 |
| | | F>30 | 1 | 1 | 0.8 | 0.5 | 0.15 | 0 |
| | | F>50 | 1 | 1 | 0.4 | 0.05 | 0 | 0 |
| | 2000 | F>10 | 1 | 1 | 1 | 1 | 1 | 0.4 |
| | | F>30 | 1 | 1 | 1 | 0.9 | 0.9 | 0 |
| | | F>50 | 1 | 1 | 1 | 0.4 | 0.3 | 0 |

*The total number of noise SNPs incorrectly identified as major IVs out of the 295 noise SNPs.

to recognize SNP 1 and 2 as major IVs in conditions characterized by low sample sizes and heritability. A threshold of $F > 30$ emerges as a balanced criterion for defining major IVs, effectively mitigating the aforementioned issues. Thus, we propose to use a partial $F > 30$ threshold in the selection of major IVs.

**Comparison with 2SLS and LIML.** We compared the proposed MR-SPLIT with the widely-used 2SLS approach and the LIML method which is particularly designed to address the weak instruments bias issue. We simulated 300 SNPs independently and randomly selected 5 SNPs as the IVs to generate the exposure variable $X$. We set $h^2 = \{0.15, 0.3, 0.5\}$ which respectively represent weak, moderate, and strong overall effect, and $\rho = (0.1, 0.2)$ where $\rho = cor(\varepsilon_{xi}, \varepsilon_{yi})$ controls the unknown confounding effect.

We set the sample size ($N$) to 1000. To ensure a fair comparison with 2SLS and LIML, we only split the sample once (i.e., no multiple splitting). We then used one subset for selecting the IVs and incorporated the other subset with the selected IVs for estimation. Both 2SLS and LIML followed the same process but did not differentiate between major and weak IVs for further causal inference. To check the impact of selection bias for 2SLS and LIML, we also did the analysis using the whole dataset for both IV selection and causal effect estimation. The

**Table 2. Simulation comparison between M\* (MR-SPLIT), LIML and 2SLS.**

| $h^2$ | $\rho$ | $\beta$ | Bias($|\beta - \hat\beta| \times 100$) | | | Est. SE | | | CP\* | | |
|---|---|---|---|---|---|---|---|---|---|---|---|
| | | | M\* | LIML | 2SLS | M\* | LIML | 2SLS | M\* | LIML | 2SLS |
| 0.15 | 0.1 | -0.08 | 0.18 | 0.17 | 4.86 | 0.1252 | 0.1776 | 0.0788 | 0.955 | 0.827 | 0.895 |
| | | 0.08 | 0.82 | 0.23 | 5.05 | 0.1253 | 0.1884 | 0.0795 | 0.957 | 0.832 | 0.886 |
| | 0.2 | -0.08 | 0.17 | 1.19 | 9.45 | 0.1230 | 0.1737 | 0.0782 | 0.949 | 0.843 | 0.740 |
| | | 0.08 | 0.31 | 1.01 | 9.44 | 0.1320 | 0.1770 | 0.0801 | 0.952 | 0.825 | 0.732 |
| 0.30 | 0.1 | -0.08 | 0.02 | 0.11 | 2.4 | 0.0520 | 0.0844 | 0.0605 | 0.958 | 0.895 | 0.929 |
| | | 0.08 | 0.13 | 0.59 | 2.94 | 0.0515 | 0.0831 | 0.0600 | 0.959 | 0.898 | 0.919 |
| | 0.2 | -0.08 | 0.43 | 0.67 | 4.77 | 0.0501 | 0.0840 | 0.0612 | 0.946 | 0.904 | 0.837 |
| | | 0.08 | 0.47 | 0.31 | 5.03 | 0.0524 | 0.0840 | 0.0621 | 0.947 | 0.905 | 0.845 |
| 0.50 | 0.1 | -0.08 | 0.32 | 0.08 | 1.12 | 0.0329 | 0.0482 | 0.0430 | 0.938 | 0.943 | 0.945 |
| | | 0.08 | 0.11 | 0.22 | 0.82 | 0.0328 | 0.0474 | 0.0424 | 0.948 | 0.931 | 0.944 |
| | 0.2 | -0.08 | 0.14 | 0.02 | 2.08 | 0.0335 | 0.0513 | 0.0457 | 0.942 | 0.909 | 0.902 |
| | | 0.08 | 0.2 | 0.03 | 2.07 | 0.0318 | 0.0469 | 0.0423 | 0.954 | 0.938 | 0.924 |

CP\*=coverage probability

simulation settings are the same as what we previously described. The only difference is that we do not split the sample and use the whole sample to do the IV selection and estimation. Results for this analysis were given in S1 Text. The respective boxplots, illustrating the distribution of estimations across 1000 simulation iterations, are provided in Figs B, C and D in S1 Text. It is evident from the results that using the entire sample for both IV selection and effect estimation results in estimates with smaller variance but larger bias, leading to a significantly higher type I error rate. In the following, we only show the results based on sample splitting.

Table 2 presents a comparative analysis of the estimation accuracy among MR-SPLIT, LIML, and 2SLS. It shows that MR-SPLIT can provide estimates with a significantly small bias. In contrast, the estimates from 2SLS exhibit large bias, especially under weak IV and substantial confounding effects (e.g., $\rho$ = 0.2). In some of the cases, LIML gives a smaller bias than MR-SPLIT does, but it has consistently larger variance than MR-SPLIT, leading to a conservative coverage probability (CP) compared to MR-SPLIT. The variance of 2SLS is uniformly smaller than the other two methods. However, given its large bias, it has the most poor coverage probability among the three methods. On the other hand, MR-SPLIT shows consistently good coverage probabilities under different scenarios, showcasing its robust performance under different conditions.

Fig 2 shows the results of the type I error of the three methods. We can observe that MR-SPLIT can effectively control Type I errors, even in the presence of strong unknown confounding. As depicted in Fig 2, both LIML and 2SLS methods exhibit much poorer performance than MR-SPLIT. Notably, 2SLS suffers from poor type I error control when the confounding effect is strong (i.e., $\rho$ = 0.2), leading to inflated error rates. LIML has high false positive rates when the SNP effects are weak (i.e., weak instruments with low $h^2$), especially under $\rho$ = 0.2. As the SNP effects increase, its performance improves; however, it can only effectively control type I errors when the instrumental variables are strong, as demonstrated in the scenario with $h^2$ = 0.5. Conversely, MR-SPLIT consistently demonstrates robust type I error control under all conditions, even under $h^2$ = 0.15 and $\rho$ = 0.2, where 2SLS and LIML exhibit their poorest performance. The inflated type I error rates lead to inflated statistical power for 2SLS and LIML. Consequently, comparing power between MR-SPLIT and these two methods may not be a fair comparison; thus, we did not show the detailed power

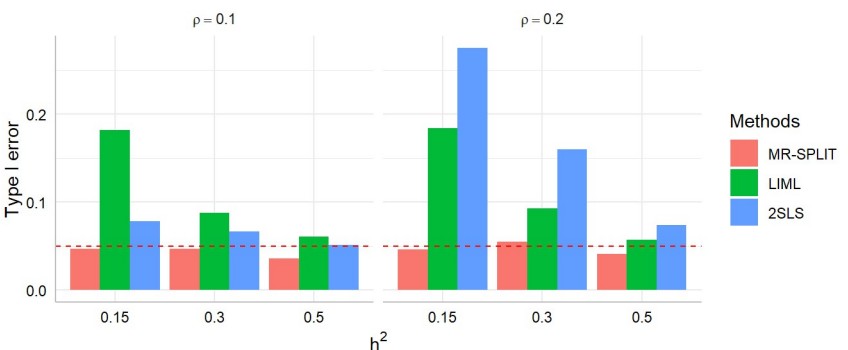

**Fig 2. Type I error comparison between MR-SPLIT, 2SLS and LIML.** The horizontal dashed line denotes the 0.05 level.

comparison here. Nevertheless, in the scenario where $h^2 = 0.5$ and $\rho = 0.1$, MR-SPLIT still attains the highest power, reaching 0.683 compared to 0.545 for 2SLS and 0.419 for LIML.

**Comparison with CFMR.**   We compared our method with CFMR under different simulation scenarios. To ensure a fair comparison with CFMR, we applied 10-fold CFMR as recommended in the CFMR work, and 2-fold MR-SPLIT with 50 random sample splits. We applied the same procedure for selecting IVs. While CFMR combined all the selected IVs into a single composite one, our method differentiated between major and weak IVs using the partial $F > 30$ criterion and only weak IVs were combined into a composite one. We also followed the simulation settings described in the CFMR work to ensure a fair comparison. We generated a set of 300 SNPs, and the minor allele frequency is fixed as 0.3 for all the SNPs. We randomly chose 5 SNP IVs to generate the exposure variable with the model $X = \sum_{j=1}^{5} G_j \alpha_j + \varepsilon_x$, and the outcome with the model $Y = X\beta + \varepsilon_y$, where

$$\begin{pmatrix} \varepsilon_x \\ \varepsilon_y \end{pmatrix} \sim N\left(0, \ 5\begin{pmatrix} 1 & 0.16 \\ 0.16 & 1 \end{pmatrix}\right)$$

We set two scenarios to comprehensively compare MR-SPLIT and CFMR:

- Scenario I: The effect sizes of the 5 SNPs are different, i.e., $\alpha = (0.4, 0.4, 0.1, 0.05, 0.05)$. Potentially, SNPs with the effect of 0.4 can be regarded as major IVs and the rest can be considered as weak ones. This also depends on the SNP heritability level $h^2$.

- Scenario II: The effect sizes of the 5 SNPs are the same, i.e., $\alpha = (0.2, 0.2, 0.2, 0.2, 0.2)$. In this case, differentiating between major and weak IVs can be challenging, presenting a less favorable condition for our method.

In each scenario, we compared the two methods in different aspects by changing the sample size ($N = \{1000, 3000, 5000\}$), variation in the exposure explained by the SNP IVs ($h^2 = \{0.15, 0.2, 0.3\}$) and the exposure's effect size for $\beta$.

Fig 3 depicts the type I error control of the two methods in scenario I and scenario II. In general, the control of type I error for the two methods is highly comparable across different settings characterized by distinct sample sizes and SNP heritability levels. Though the type I error is a little inflated for MR-SPLIT under a small sample size ($N = 1000$), particularly in scenario II, it controls the type I error well as the sample size increases.

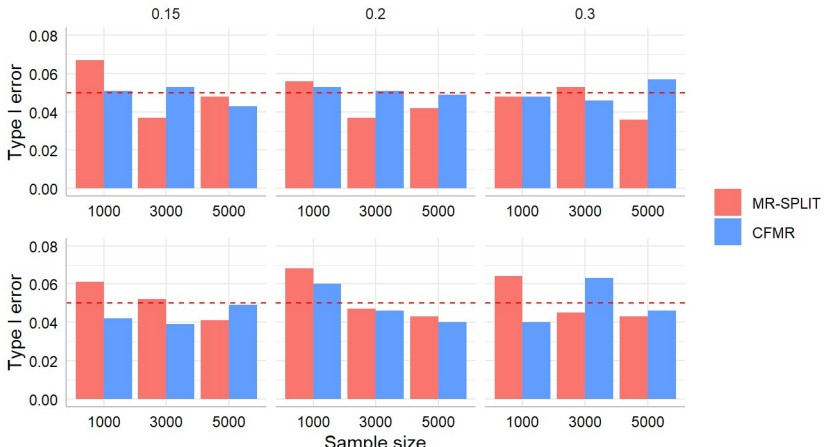

**Fig 3. Comparison of type I error between MR-SPLIT and CFMR in Scenario I (top) and II (bottom).** The horizontal dashed line denotes the 0.05 level.

Fig 4 shows the results of the power for the two methods in these two scenarios. Regardless of the settings, MR-SPLIT consistently exhibits higher power than CFMR. This discrepancy becomes especially noticeable when the IVs are relatively weak (i.e., $h^2 = 0.15$).

In S1 Text, we also presented the estimation performance of both methods when $\beta = 0$ and 0.08 in Figs E-J. The results reveal minimal difference in the causal effect estimation between the two methods. However, a noticeable distinction is the smaller standard error observed in MR-SPLIT across nearly all the scenarios, resulting in a smaller Root Mean Square Error (RMSE) (see Fig K in S1 Text) and higher statistical power when compared to CFMR. This aligns well with the theoretical finding in Theorem 1, though the result is proved under a 2-fold sample split. The findings further underscore the advantages of MR-SPLIT.

In summary, MR-SPLIT consistently demonstrates robust type I error control when compared to 2SLS and LIML across various simulation settings. In comparison to CFMR, MR-SPLIT exhibits superior performance, yielding smaller RMSE and higher statistical power. Even under a less favorable condition for MR-SPLIT, the type I error can be controlled when the sample size is reasonably large. The simulation results further corroborate our theoretical

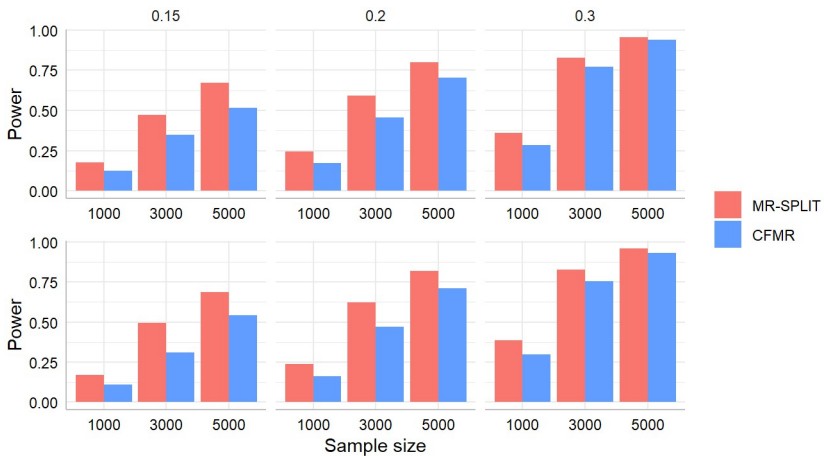

**Fig 4. Power comparison between MR-SPLIT and CFMR in Scenario I (top) and II (bottom).**

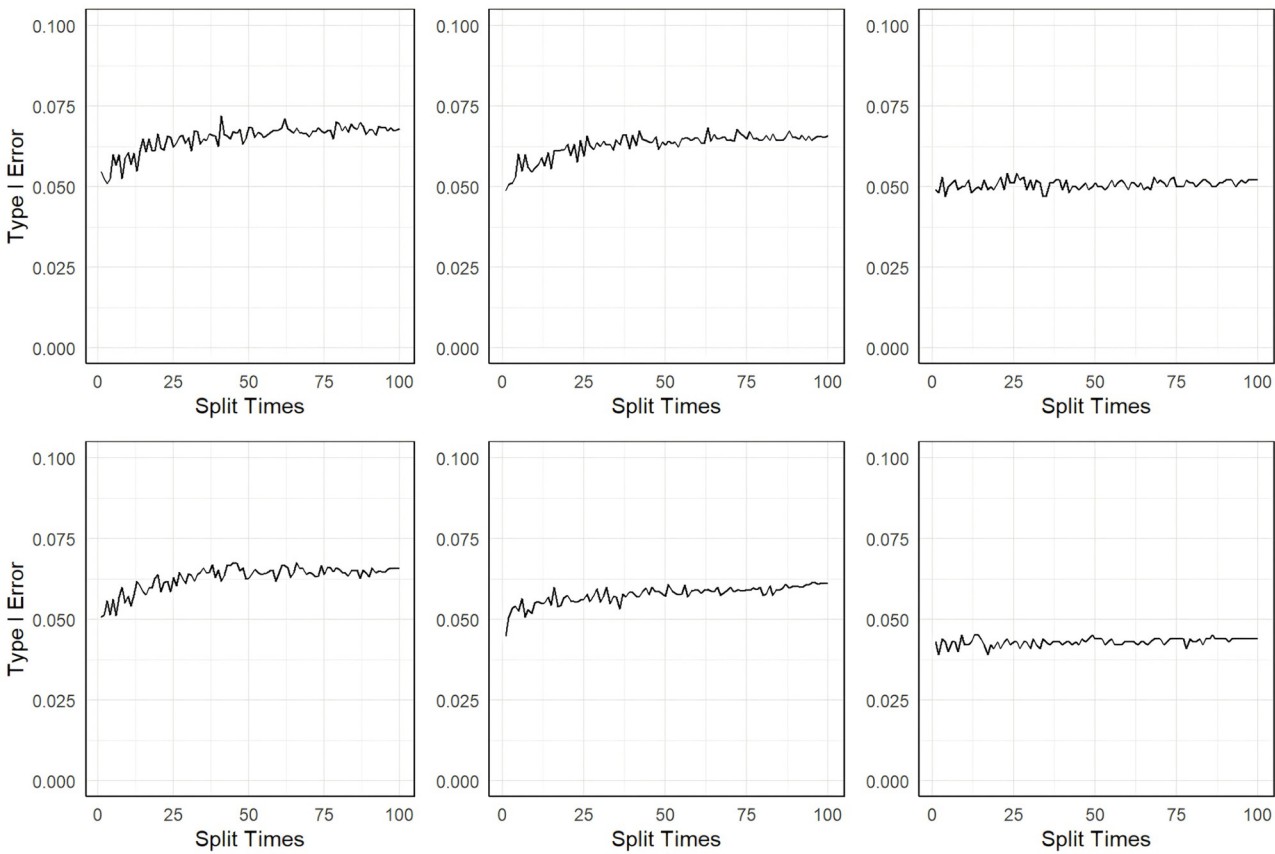

**Fig 5. Type I error under different sample sizes:** $N = 500$(left), 1000(middle), 2000(right), and under different $h^2$: 0.15 (top) and 0.2 (bottom).

finding, consistently showing that MR-SPLIT results in smaller standard errors for causal effect estimation compared to CFMR, which leads to higher statistical power when testing for the causal effect.

**Evaluation of Multiple data splitting to improve robustness.** Intuitively, more data splitting should yield more robust results, which, however, would entail higher computational resource usage. We implemented our methods under different splitting times, different sample size $N$ and different $h^2$ values, to check if we can find an efficient number of splitting. In our simulations, the true causal effect of the exposure on the outcome is set to equal 0.2 ($\beta = 0.2$), and the sample size ranges from 500 to 2000 ($N = 500, 1000, 2000$). We did simulations in Scenario I as described in section Comparison with CFMR. Fig 5 demonstrates how the type I error fluctuates with an increasing number of splits. To obtain a smoother estimate of the type I error rate, we repeated the simulation 5,000 times Under a small sample size, the type I error rates get stable as the number of sample splits increases. Though the type I error increases as the sample split times increase under small sample sizes, this increase is considered acceptable, particularly in light of the associated boost in power (see Fig 5), which is especially pertinent for smaller sample sizes.

Fig 6 shows the empirical power under different sample sizes and $h^2$. The type I error and power results when $h^2 = 0.3$ can be found in Figs L and M in S1 Text. When the sample size is small ($N = 500$) and the IVs are relatively weak ($h^2 = 0.15$), the power gets stabilized after 50 splits. As the sample size increases, there is a decrease in the need for the number of sample

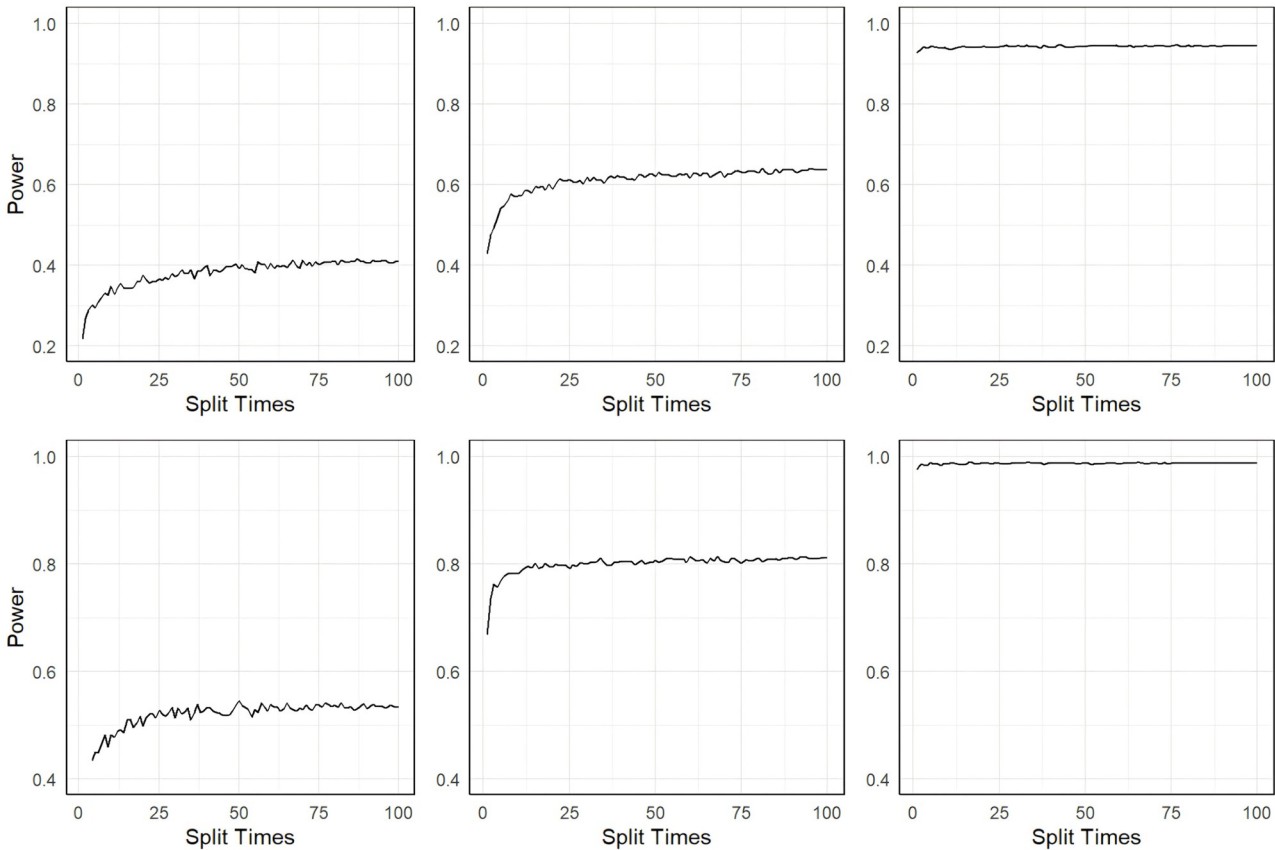

**Fig 6. Empirical power under different sample sizes:** $N = 500$(left), $1000$(middle), $2000$(right), and under different $h^2$: 0.15 (top) and 0.2 (bottom).

splits to maintain stable power. This indicates that in practical data analysis, it is possible to estimate the exposure heritability based on the selected SNP IVs, and thereafter determine the appropriate number of sample splits. In any case, opting for 50 sample splits represents a highly conservative option.

## Case study

We demonstrated the effectiveness of our method by applying it to the Chronic Renal Insufficiency Cohort (CRIC) dataset, to understand the progression of chronic kidney disease (CKD).

CKD is evaluated utilizing two straightforward tests: a blood test known as the estimated glomerular filtration rate (eGFR) and a urine test, the urine albumin-creatinine ratio (uACR). Both eGFR and uACR measure kidney function, with low eGFR and high uACR values indicating impaired kidney function. In this application, we are interested in evaluating the causal relationship between CKD and apparent Treatment-Resistant Hypertension (aTRH). aTRH is a condition where a patient's blood pressure remains above target levels despite using three different classes of antihypertensive drugs at optimal doses, typically including a diuretic. The definition of aTRH also extends to cases where four or more medications are required to effectively control blood pressure [32]. In a recent two-sample MR analysis using summary statistics, Yu et al. [33] identified the causal effect of higher kidney function (measured by eGFR estimated from serum creatinine) on lower systolic blood pressure. To date, the causal

relationship between CKD and aTRH and the causal link between them remains to be established [34–37]. To this end, we utilized eGFR and uACR as the exposure variable and aTRH as the outcome, applying MR-SPLIT for our analysis. For comparative purposes, we also employed CFMR and 2SLS on the same dataset. Given that the outcome variable (aTRH) is binary (0/1) in nature, the LIML method is not suitable in this analysis.

**Genetic data processing.** The original data have 3,541 samples containing 970,342 SNPs. Our initial step involved removing SNPs with missing rate larger than 10%, resulting in 886,384 SNPs. After excluding SNPs with minor allele frequency (MAF) lower than 0.05, 762,664 SNPs were left. The next phase entailed the elimination of SNPs with p-values less than 1e-5 in the Hardy-Weinberg equilibrium test, which narrowed our SNP count down to 693,848. To ensure the robustness of our genetic instruments, we then implemented LD pruning. SNPs were filtered out in close LD by considering pairs of SNPs within a window of 100 kb. If a pair of SNPs has an LD measure ($r^2$) exceeding 0.64, one SNP from the pair is removed. After completing all these steps, we were left with 467,597 SNPs.

**Causal effect of eGFR on aTRH.** In the initial dataset, eGFR values were obtained on multiple occasions. For consistency and relevance, we selected the eGFR measurements corresponding to visit number 3, which also represents the baseline assessment. Following the exclusion of samples with missing values for either eGFR or aTRH, and then combined with the SNP data, our analysis proceeded with a total of $N = 1,353$ samples. A simple logistic regression shows there is a strong association between aTRH and eGFR ($p < 2 \times 10^{-16}$). We would like to evaluate if this association is causal. Fig S20 shows the boxplots of eGFR in aTRH positive and negative groups.

Next, we proceeded with the MR-SPLIT and used SIS for preliminary screening, reducing the number of SNPs from ultra-high to high. To optimize computational efficiency in the analysis, we first conducted univariate regression of each SNP against the exposure before applying sample split, using the whole data set. A total of 4,580 SNPs ($p < 0.01$) remained for further analysis. The removed SNPs would most likely be screened out by the SIS procedure in subsequent steps even after the sample split if not discarded at this stage. For each of the 50 sample splits, we used the 'screening' function from the R package 'screening' with the SIS option. The number of SNP variables retained post-screening adhered to the default setting, which is half the size of the sample. In this real data analysis, instead of applying the LASSO algorithm to select and estimate SNP effects, we employed a high-dimensional inference procedure, specifically a LASSO-projection method which provides debiased coefficient estimates and hence a valid p-value for each coefficient. This is done by using the 'lasso.proj' function in the R 'hdi' package [30]. As the regular LASSO estimates are biased, this approach can give debiased estimates and further provide p-values for testing each coefficient. To compare the performance of the LASSO-projection with the regular LASSO, We conducted a simulation (detailed in Section 6 in S1 Text). The results show that the LASSO-projection method slightly outperforms LASSO, exhibiting higher power and better control of the type I error rate and smaller RMSE. After getting the p-values for each SNP, we retained those with a p-value less than or equal to 0.05. This resulted in an average of 98 IVs out of 50 sample splits. We used the partial $F > 30$ as the criterion to declare major IVs. And the weak IVs were then combined into a composite IV. Finally, we used both the composite IV and the major IV(s) to obtain the causal effect estimate and the p-value.

Fig 7 shows the p-value distribution and the causal effect estimates out of 50 sample splits. The majority of p-values obtained from MR-SPLIT are below 0.05, and the majority of the estimated causal effects $\hat{\beta}$ is centered around -0.0343 (indicated by the black dashed line). In these 50 sample splits, there was an average of 98.06 IVs incorporated into the model for the causal

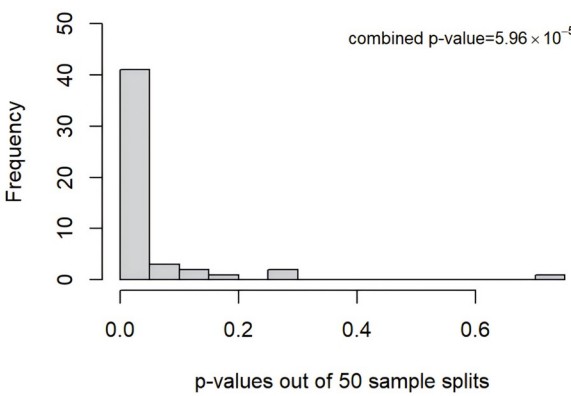

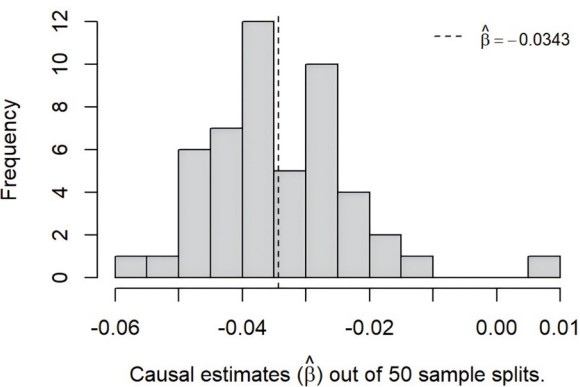

**Fig 7. Histogram of p-values and causal effect estimates from 50 sample splits when eGFR is treated as the exposure.**

effect estimate and the majority were classified as weak IVs. Among these, an average of 0.54 IVs were identified as major IVs each time. After aggregating all the results using Cauchy's combination rule, our method provided an estimate of $\hat{\beta} = -0.0343$ (OR = 0.9663), with an aggregated p-value of 5.96 #x00D7; $10^{-5}$. We also tried lowering the partial F threshold to 20, which yielded slightly more major IVs than the $F > 30$ threshold (see Fig V in S1 Text). Among the 50 sample splits, an average of 4.26 IVs were identified as major IVs each time. The results show that the p-value for MR-SPLIT improved slightly (from $5.9 \times 10^{-5}$ to $2.9 \times 10^{-6}$), but the estimates remained nearly the same ($\hat{\beta} = -0.0342$).

We also applied the CFMR method with a 10-fold split. The CFMR method yielded an average estimate of $\hat{\beta} = -0.0378$ (OR = 0.9629), with a p-value of $< 1 \times 10^{-5}$. For reference, simply conducting the 2SLS method yields an estimate of $\hat{\beta} = -0.0407$ (OR = 0.9601), with a p-value of $< 1 \times 10^{-7}$. The three methods established a consistent causal relationship between eGFR and aTRH.

**Causal effect of uACR on aTRH.** Following a similar procedure, we excluded samples with incomplete data for either uACR or aTRH. After merging the remaining data with the SNP data, the dataset was reduced to 1,324 samples. The distribution of uACR is very skewed (to the right) (See Fig W in S1 Text). We opted to do a logarithmic transformation of uACR, denoted as log(uACR). A simple logistic regression shows there is a strong association between log(uACR) and aTRH ($p = 4.6 \times 10^{-12}$). Similar procedures as described before were followed for further analysis.

Fig 8 shows the p-value distribution as well as the causal effect estimate out of 50 sample splits with MR-SPLIT. In these 50 sample splits, there were on average 74.3 SNPs selected as IVs with the majority as weak ones for the causal effect estimate. Among them, an average 0.22 IVs were identified as major IV each time. After aggregating all the results, the final causal estimate was $\hat{\beta} = 0.1675$ (OR = 1.186), with a p-value of $1.9 \times 10^{-3}$. For comparison, CFMR provided an estimate of $\hat{\beta} = 0.1584$ (OR = 1.1716), with a p-value of $5.2 \times 10^{-5}$. The two methods yielded statistically significant results and presented comparable estimates. While applying 2SLS on the same dataset, we also observed significant results (p-value = $1.3 \times 10^{-5}$), albeit with a different causal effect estimate of $\hat{\beta} = 0.0363$.

Integrating the results from the two analyses that utilized eGFR and uACR separately as exposures, we infer that there exists a causal relationship between CKD function and aTRH. Specifically, a lower eGFR and a higher uACR tend to contribute to an increased risk of aTRH. However, we recognize the limited sample size of this study, which necessitates cautious

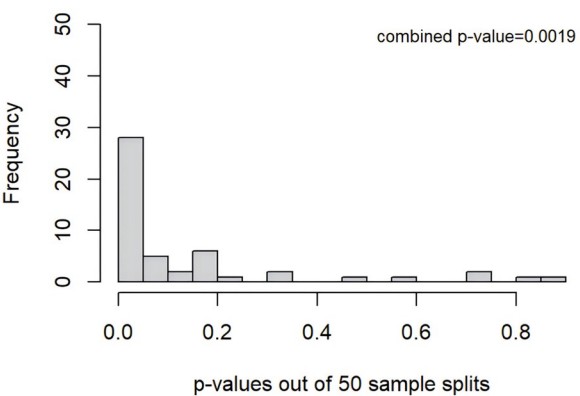
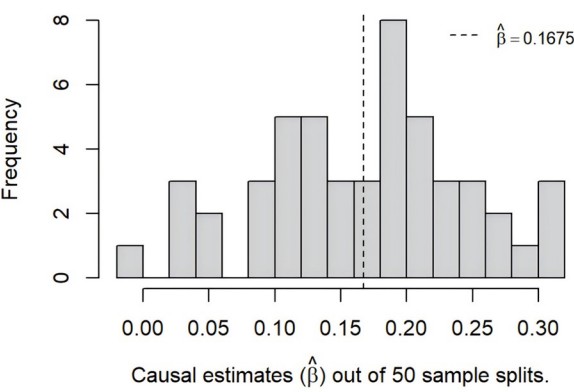

**Fig 8. Histogram of p-values and causal effect estimates from 50 sample splits when log(uACR) is treated as the exposure.**

interpretation of the causal relationship identified. To assess the possibility of a reverse causal effect, we require a method capable of accommodating a binary exposure variable, such as aTRH in this context. This will be explored in our future studies.

## Discussion

MR analysis has been an instrumental means in epidemiology studies, enabling the assessment and revelation of causal connections between exposures or interventions and particular outcomes, leveraging genetic variants as IVs to mitigate confounding factors. In this study, we introduced an innovative adaptive sample splitting method known as MR-SPLIT, designed to address the issue of IV selection bias and weak instruments in the context of one-sample MR analysis using individual-level data. By a random sample split, we use half sample to select IVs and another independent half to estimate the causal effect, hence avoiding the winner's curse problem by using the same data for IV selection and causal effect estimation. Additionally, we presented a multi-sample splitting strategy to further enhance the robustness of causal estimation and testing. Our approach involves the adaptive identification of major and weak IVs and further aggregate weak IVs to form a composite IV. The final set of IVs comprises the major IV(s) and the composite IV. Such a strategy, as shown in the theoretical evaluation and simulation results, yields a more efficient causal estimate than CFMR, thereby enhancing testing power. In addition, MR-SPLIT shows consistently superior performance in terms of coverage probability. Therefore, MR-SPLIT offers significant improvements over existing methods by effectively handling weak instruments in one-sample MR analysis and providing robust results with enhanced statistical power.

In comparison to the traditional 2SLS and LIML methods, MR-SPLIT yields less biased results and effectively controls type I error, under different simulation settings. Compared to the CFMR approach, which is designed to tackle weak IV issues, our approach provides estimates with smaller variance and higher statistical power. In the application to the CRIC dataset, both MR-SPLIT and CFMR produce highly comparable results. We established the causal impact of kidney function, as assessed by eGFR and uACR, on aTRH. It is worth noting that both CFMR and MR-SPLIT not only address the issue of weak instrument bias (i.e. finite-sample bias from IV analysis with a given set of IVs), they also solve the problem of "winner's curse" (i.e. bias due to variant selection in the same dataset as the analysis is performed, in particular under a high-dimensional scenario). The two sets of biases are related but are conceptually distinct. By employing sample splitting strategies, both methods tackle the two bias issues

and offer a solution to one-sample MR analysis. On the other hand, as shown in our theoretical evaluation as well as the intensive simulation studies, MR-SPLIT demonstrates superior performance compared to CFMR. Within the proposed sample splitting strategy, additional tasks such as nonlinear causal estimation can also be executed using one-sample individual-level data.

In the process of selecting IVs, CFMR recommends employing predictive methodologies, such as LASSO regression, for their efficacy in enhancing prediction accuracy through variance minimization. However, this approach often introduces bias in effect estimates, as it may incorporate SNPs without significant association with the exposure—potentially compromising the relevance assumption for IVs. On the other hand, 2SLS analysis prioritizes the use of predicted exposure values in its secondary causal inference phase, underlining the importance of prediction accuracy for causal estimation. Recent advancements in the realm of high-dimensional statistical inference offer a promising solution by enabling the evaluation of estimation uncertainty for LASSO-derived estimates [30]. This is achieved through a de-biasing step that facilitates the calculation of p-values, thereby presenting an innovative approach for SNP IV selection within the context of high-dimensional SNP-exposure regressions. This technique allows for the derivation of p-values for individual SNPs, enabling the validation of IV suitability through a p-value based method. Unlike traditional practices that determine p-values by fitting each SNP individually in marginal regressions, this approach fits all SNPs (after the SIS step) in a multiple regression model. This yields partial SNP effect estimates and hence partial p-values, offering a nuanced perspective compared to conventional methods. By adopting a p-value threshold criterion (e.g., $p < 0.05$), the selected SNPs meet the relevance assumption, providing a more robust framework for IV selection. In our analysis, we observed that the LASSO variable selection technique typically identifies a greater number of IVs compared to the debiased LASSO method. If computational resources are not a limiting factor, we recommend the implementation of the debiased LASSO approach in practical applications.

While MR-SPLIT offers notable advantages, there is still considerable potential for further enhancement and refinement. In this work, we applied the partial $F$ statistics for identifying major IVs, which does not rule out the application of other measures such as those studied in James and Motohiro [27]. Any statistical measure capable of ranking the effect sizes of the selected IVs could be considered for enhancing the robustness and effectiveness of our approach. It is essential to devise robust methods for discerning between major and weak IVs. This represents a promising direction for future research. It is worth mentioning that we do not specify the ratio of major IVs to weak IVs; their quantities are entirely determined by the data itself, that is, based on the strength calculated from the IVs. On the other hand, as revealed by the simulation studies, the declaration of major IVs may vary under different F thresholds and under different sample sizes and SNP heritability levels. In real applications, the $F > 30$ threshold can be relaxed under a small sample size and low heritability level. The genomewide SNP heritability can be estimated with software such as GCTA [38].

In addition, we employed a straightforward weighted combination approach to aggregate the information from all weak IVs into a single composite IV. Other advanced machine learning techniques could also be borrowed by minimizing information loss which could potentially yield improved results.

An additional constraint of our approach is that we did not take the pleiotropic effects into consideration. Addressing pleiotropy can be a complex endeavor, but there are several test statistics available to identify its presence [39, 40]. Integrating these statistics into our method presents a challenge, as it requires the repeated selection of new IVs at each sample split and the detection of pleiotropic effects associated with the chosen IVs. We plan to tackle this issue in our future investigation. Studies also show that incorporating the invalid IVs with

uncorrelated and correlated horizontal pleiotropic effects can potentially increase power and decrease bias [41–43]. We will investigate this in our future work.

The concept of sample splitting and cross-fitting instruments introduced in this study has potential applications beyond the scope of traditional one-sample MR analyses using individual-level data. For example, this framework can be adapted for use in multiple exposure MR analyses, where it would involve adapting the existing approach to handle multiple sets of selected IVs simultaneously. For another example, the proposed framework enables the investigation of potential non-linear causal relationships through a control function approach while effectively addressing the two bias issues previously mentioned. Accomplishing this task is not feasible with summary statistics, highlighting the framework's capability to provide more nuanced insights into causal mechanisms that cannot be captured by summary-level data. In essence, the expansion of our methodology to encompass various types of MR analyses could facilitate innovative research into causal relationships, opening new avenues for investigation.

## Supporting information

**S1 Text. Supplementary Materials for "MR-SPLIT: A novel method to address selection and weak instrument bias in one-sample Mendelian randomization studies".**
(PDF)

## Acknowledgments

The Chronic Renal Insufficiency Cohort (CRIC) Study was conducted by the CRIC Study Investigators and supported by the National Institute of Diabetes and Digestive and Kidney Diseases (NIDDK). This manuscript was not prepared in collaboration with Investigators of the CRIC Study and does not necessarily reflect the opinions or views of the CRIC Study, or the NIDDK.

## Author Contributions

**Conceptualization:** Yuehua Cui.

**Data curation:** Ruxin Shi.

**Formal analysis:** Ruxin Shi, Ling Wang, Stephen Burgess.

**Investigation:** Ruxin Shi, Stephen Burgess, Yuehua Cui.

**Methodology:** Ruxin Shi, Yuehua Cui.

**Project administration:** Yuehua Cui.

**Resources:** Ling Wang, Yuehua Cui.

**Software:** Ruxin Shi.

**Supervision:** Yuehua Cui.

**Validation:** Ruxin Shi.

**Visualization:** Ruxin Shi.

**Writing – original draft:** Ruxin Shi, Yuehua Cui.

**Writing – review & editing:** Ruxin Shi, Yuehua Cui.

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
