## [Decision Letter · Decision Letter 0]

4 Apr 2024

Dear Dr Cui,

Thank you very much for submitting your Research Article entitled 'MR-SPLIT: a novel method to address selection and weak instrument bias in one-sample Mendelian randomization studies' to PLOS Genetics.

The manuscript was fully evaluated at the editorial level and by independent peer reviewers. The reviewers appreciated the attention to an important topic but identified some concerns that we ask you address in a revised manuscript.

We therefore ask you to modify the manuscript according to the review recommendations. Your revisions should address the specific points made by each reviewer.

Yours sincerely,

Jean Morrison

Guest Editor

PLOS Genetics

Xiaofeng Zhu

Section Editor

PLOS Genetics

Reviewer's Responses to Questions

**Comments to the Authors:**

Reviewer #1: The manuscript presented, MR-SPLIT, a novel method to address selection and weak instrument bias in one-sample MR analysis. The authors first provided the theoretical justification in terms of the efficiency for the causal effect estimate, and then conducted various simulations, parallelized with real data applications, to illustrate the advantage of MR-SPILIT over other competing methods. Overall, the manuscript is of great interest and well-written, I just have the following comments for further improvement.

1: Given that including more correlated IVs, rather than only independent IVs through LD clumping or pruning, can increase the power of MR analysis. I wonder whether MR-SPLIT can be developed for correlated IVs. I suggested the authors to discuss this.

2. It seems that there is no gold standard to determine either the major or the weak IVs. For fixed number of IVs, dose the performance of MR-SPLIT rely on the ratio of the number of major IVs to the number of weak IVs.

3: In the real data analysis, it seems that the power advantage of MR-SPLIT over CFMR is not that obvious, and CFMR can even produce much smaller p value than MR-SPLIT. I suggested the authors to investigate the possible reasons. For example, are there any IV associated with the outcome? If so, what happened if we remove them from the analysis. Indeed, this can partly alleviate the issue of pleiotropy.

4. In the real data analysis, what is the criteria to determine the major IVs or the weak IVs? Again, what happened if we changed the ratio of the number of major IVs to the number of weak IVs.

5. There are some typos in the manuscript, For example, in line 4 of the 4.3 section, p=4.6*10(-12). I suggested the authors to double check the whole manuscript again.

Reviewer #2: This manuscript proposed a novel method named as Mendelian Randomization with adaptive Sample-sPLitting with cross-fitting InstrumenTs (MR-SPLIT) to address IV selection issue and weak instrument bias under the 2SLS IV regression framework.

Overall, the paper is well-written and the methods are nice outlined both in the main text and in supplementary materials. Below are my comments and questions:

Introduction:

P2. The author explained the assumptions of IVs in the MR analysis. The violation of the relevance assumption leads to a 'weak instrument' issue. Please describe why the challenge of IV selection should also be considered when this problem was first mentioned.

P5 and P7. The author only introduced methods that handled the weak instrument issue. Are there any methods that consider the IV selection issue? Please add some citations.

P8. Please summarize how this method addressed the bias of IV selection.

Methods:

In section 2.1.1 P3. Please describe the definition of partial F-statistics and explain why to use partial F-statistics as criteria.

In section 2.1.2. Please explain how to use I_1 and I_2 to select major and weak IVs, respectively. Could you please explain whether the weak IVs used to construct composite IV are the same as the IVs obtained in sample I_1?

In Remark 2. The author described that MR-SPLIT benefits from a 2-fold sample splitting. It selects and separates major and weak IVs in I_1, then forms the cross-fitting composite IV in I_2. Please explain why 2-fold sample splitting can deal with the weak instrument bias and IV selection issue.

In section 2.1.5 algorithm. How to define the value of L random split in the simulations studies and real data analysis?

Simulation Study:

In section 3.1. The author can determine what criteria to distinguish the major and weak IVs by conducting simulations. Please describe the value of the threshold used in the real data analysis and how to determine this threshold.

In sections 3.1 and 3.2. Please add more details to explain how to randomly generate 300 SNPs. Do you generate the simulated datasets following the model (1) in section 2? Are these SNPs independent with each other? Do you consider the LD matrix when generating the SNPs?

P2. Please add more details on conducting simulations to check the impact of selection bias, including specific methods for sample division.

Case Study:

P2. Please explain the reason for using a LASSO-projection method. It would be helpful to provide some numerical results to show that this one is better than LASSO algorithm to select and estimate SNP effects.

Please describe the value of the partial F statistic threshold and the reason.

Could you provide any references that confirm the causal relationship between eGFR and aTRH? If available, please include the citations.

In real data analysis, the author only compares the MR-SPLIT with CFMR. Please add more results to compare with LIML and 2SLS.

Reviewer #3: The author introduces MR-Split, an interesting extension of the work of Denault and colleagues on CFMR. The limitations of the CFMR are well presented, and the approach proposed by the author is sound and seems to address correctly the limitations of CFMR. They even provide theoretical arguments for MR split superiority compared to CFMR.

I am positive about accepting this paper, but I have a minor revision.

Comments:

1) The paper needs some polishing/sharpening. For example, at the start of section two, the authors say, “assume the following 2SLS” model. 2SLS is an estimation method for this kind of model but not a model by itself. This model is an SEM. Another example in the supplementary material on page 14 is that the authors wrote, “To prove S14, we need to show ???” This is obviously a latex mistake regarding the references. It would be good if this kind of typos/impreciseness were corrected for the next version.

2) Regarding Figure 5, I appreciate that the authors point out the lack of control of type one error when increasing the number of splits in a small sample size. I suspect that some simple bootstrapping method could be used to correct this bias. If this is not possible, I still agree that the power gain is worth a slight increase in type I error. Furthermore, it would be good to have a smoother estimate of the type I error. Could the authors perform more simulations(given that this is for a small sample, I suspect that should be doable.

**Have all data underlying the figures and results presented in the manuscript been provided?**

Reviewer #1: None

Reviewer #2: Yes

Reviewer #3: Yes

PLOS authors have the option to publish the peer review history of their article (what does this mean?). If published, this will include your full peer review and any attached files.

Reviewer #1: No

Reviewer #2: No

Reviewer #3: **Yes: **William R.P. Denault

---

## [Decision Letter · Decision Letter 1]

18 Jul 2024

Dear Dr Cui,

Thank you very much for submitting your Research Article entitled 'MR-SPLIT: a novel method to address selection and weak instrument bias in one-sample Mendelian randomization studies' to PLOS Genetics.

The manuscript was fully evaluated at the editorial level and by independent peer reviewers. The reviewers appreciated the attention to an important topic but identified some concerns that we ask you address in a revised manuscript.

We therefore ask you to modify the manuscript according to the review recommendations. Your revisions should address the specific points made by each reviewer.

To resubmit, log into your Editorial Manager account and select the option 'Revise Submission' in the 'Submissions Needing Revision' folder.

Yours sincerely,

Jean Morrison

Guest Editor

PLOS Genetics

Xiaofeng Zhu

Section Editor

PLOS Genetics

Thank you for submitting this revision. Both reviewers had positive comments. Reviewer 2 has requested some clarification of notation and methods. If these minor changes are addressed, we should be able to move forward with accepting your manuscript.

Reviewer's Responses to Questions

**Comments to the Authors:**

Reviewer #1: The authors have addressed all my comments.

Reviewer #2: Review of “MR-SPLIT: a novel method to address selection and weak instrument bias in one-sample Mendelian randomization studies” by Ruxin Shi, Ling Wang, Stephen Burgess and Yuehua Cui

Summary:

This manuscript proposed a novel method named as Mendelian Randomization with adaptive Sample-sPLitting with cross-fitting InstrumenTs (MR-SPLIT) to address IV selection issue and weak instrument bias under the 2SLS IV regression framework.

Overall, the paper is well revised, both in the main text and in supplementary materials based on the major comments, but I have the following minor suggestions:

Methods

1. Please include brief descriptions in Figure 1 to outline the workflow of MR-SPLIT and refine this flow chart. Specifically, please adjust the size of the rectangles (the third row).

2. In Sections 2.1.2 and 2.1.3, please describe the notations α ^ and {G_M,G ^_W} clearly.

3. In Section 2.1.3 Remark 1, please explain the notation n_k in G ~_(k,n_k×1).

Simulations

1. In Section 3.1.1, please add more details and explain the processes for generating independent SNPs, the exposure, and the outcome clearly (i.e., the value of β and error terms).

2. Please add the explanations of the last column (Noises) in the caption of Table 1 and Section 3.1.1.

3. In Section 3.1.2, the first line has an extra right parentheses on page 11.

**Have all data underlying the figures and results presented in the manuscript been provided?**

Reviewer #1: None

Reviewer #2: None

PLOS authors have the option to publish the peer review history of their article (what does this mean?). If published, this will include your full peer review and any attached files.

Reviewer #1: No

Reviewer #2: No

---

## [Decision Letter · Decision Letter 2]

9 Aug 2024

Dear Dr Cui,

We are pleased to inform you that your manuscript entitled "MR-SPLIT: a novel method to address selection and weak instrument bias in one-sample Mendelian randomization studies" has been editorially accepted for publication in PLOS Genetics. Congratulations!

Yours sincerely,

Jean Morrison

Guest Editor

PLOS Genetics

Xiaofeng Zhu

Section Editor

PLOS Genetics

Comments from the reviewers (if applicable):

Reviewer's Responses to Questions

**Comments to the Authors:**

Reviewer #1: I have no further comments, the manuscript are acceptable.

**Have all data underlying the figures and results presented in the manuscript been provided?**

Reviewer #1: None

PLOS authors have the option to publish the peer review history of their article (what does this mean?). If published, this will include your full peer review and any attached files.

Reviewer #1: No

**Data Deposition**

http://datadryad.org/submit?journalID=pgenetics&manu=PGENETICS-D-24-00217R2

**Press Queries**

---

## [Editor Report · Acceptance letter]

29 Aug 2024

PGENETICS-D-24-00217R2 

MR-SPLIT: a novel method to address selection and weak instrument bias in one-sample Mendelian randomization studies 

Dear Dr Cui, 

We are pleased to inform you that your manuscript entitled "MR-SPLIT: a novel method to address selection and weak instrument bias in one-sample Mendelian randomization studies" has been formally accepted for publication in PLOS Genetics! Your manuscript is now with our production department and you will be notified of the publication date in due course.

With kind regards,

Anita Estes

PLOS Genetics

On behalf of:
